# ATOKEN: A UNIFIED TOKENIZER FOR VISION

## ABSTRACT

We present ATOKEN, the first unified visual tokenizer that achieves both high-fidelity reconstruction and semantic understanding across images, videos, and 3D assets. Unlike existing tokenizers that specialize in either reconstruction or understanding for single modalities, ATOKEN encodes these diverse visual inputs into a shared 4D latent space, unifying both tasks and modalities in a single framework. Specifically, we introduce a pure transformer architecture with 4D rotary position embeddings to process visual inputs of arbitrary resolutions and temporal durations. To ensure stable training, we introduce an adversarial-free training objective that combines perceptual and Gram matrix losses, achieving state-of-the-art reconstruction quality. By employing a progressive training curriculum, ATOKEN gradually expands from single images, videos, and 3D, and supports both continuous and discrete latent tokens. ATOKEN achieves 0.21 rFID with 82.2% ImageNet accuracy for images, 3.01 rFVD with 40.2% MSRVTT retrieval for videos, and 28.19 PSNR with 90.9% classification accuracy for 3D. In downstream applications, ATOKEN enables both visual generation tasks (*e.g.*, image generation with continuous and discrete tokens, text-to-video generation, image-to-3D synthesis) and understanding tasks (*e.g.*, multimodal LLMs), achieving competitive performance across all benchmarks. These results shed light on the next-generation multimodal AI systems built upon unified visual tokenization.

## 1 INTRODUCTION

Large Language Models (LLMs) (Chowdhery et al., 2023; Achiam et al., 2023; Touvron et al., 2023; Team et al., 2023; Guo et al., 2025) have achieved unprecedented generalization, with single models handling coding, reasoning, translation, and numerous other tasks that previously required specialized systems. This versatility largely stems from transformer architectures and simple tokenizers, such as BPE (Sennrich et al., 2015), which convert all text types – code, documents, tables, and multiple languages – into a unified token space. This shared representation enables efficient scaling and seamless knowledge transfer across language tasks.

In contrast, visual representations remain fragmented due to inherent complexities. Unlike text's discrete symbolic nature, visual tasks demand distinct levels of abstraction: generation requires tokenizers that preserve low-level visual details for reconstruction, while understanding requires encoders that extract high-level semantic features through text alignment. Moreover, visual data exists in disparate formats: 2D grids for images, temporal sequences for videos, and varied 3D representations (*e.g.*, meshes, voxels, and Gaussian splats) (Mescheder et al., 2019; Achlioptas et al., 2018; Mildenhall et al., 2021). Without a shared representation, vision systems remain limited, unable to achieve the generalization and transfer learning that characterizes modern language models.

Despite recent progress, unified visual tokenizers face three fundamental challenges. First, existing approaches optimize for either reconstruction or understanding, but not both: visual encoders (Radford et al., 2021; Zhai et al., 2023; Bolya et al., 2025) achieve semantic alignment but lack pixel-level detail, while VAE-based tokenizers (Esser et al., 2020; Rombach et al., 2022; Polyak et al., 2024; Yu et al., 2022b) preserve visual details but lack semantic understanding. Second, architectural choices create different limitations: convolutional tokenizers exhibit diminishing returns when scaling model parameters (Xiong et al., 2025), while transformer tokenizers (Yu et al., 2021; Wang et al., 2024b; Hansen-Estruch et al., 2025) achieve better scaling but suffer from severe adver-

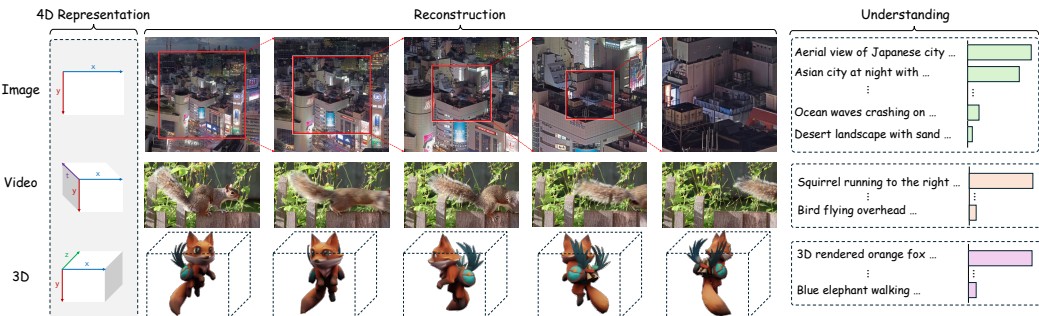

Figure 1: **AToken on images, videos, and 3D.** Our method uses a shared 4D latent space (left) to produce high-fidelity reconstructions (middle: zoomed regions for images, temporal frames for videos, viewpoints for 3D) while preserving semantic understanding (right: zero-shot text retrieval).

sarial training instabilities. Third, recent unification efforts remain limited to images (Deng et al., 2025; Wu et al., 2024c; Ma et al., 2025), while video and 3D modalities remain unexplored.

We present ATOKEN, a general-purpose visual tokenizer that achieves *high-fidelity reconstruction* and *rich semantic understanding* across *images*, *videos*, and *3D*. Our model learns a unified representation that captures both fine-grained visual details and high-level semantics, accessible through progressive encoding: semantic embeddings for understanding, low-dimensional continuous latents for generation, and discrete tokens via quantization. This design enables the next generation of multimodal systems that seamlessly handle both understanding and generation across visual modalities.

To address format discrepancies across visual modalities, we introduce a sparse 4D representation where each modality naturally occupies different subspaces: images as 2D slices, videos as temporal stacks, and 3D assets as surface voxels extracted from multi-view renderings (Xiang et al., 2024). We implement this through a pure transformer architecture with space-time patch embeddings and 4D Rotary Position Embeddings (RoPE), enabling efficient scaling and joint modeling across all modalities while maintaining native resolution and temporal length processing.

To overcome training instabilities that affect transformer-based visual tokenizers, we develop an adversarial-free loss combining perceptual and Gram matrix terms. This approach achieves state-of-the-art reconstruction quality while maintaining stable, scalable training. We introduce a progressive curriculum that builds capabilities incrementally: starting from a pretrained vision encoder, jointly optimizing reconstruction and understanding for images, extending to videos and 3D data, with optional quantization for discrete tokens. This curriculum reveals that multimodal training can enhance rather than compromise single-modality performance – our final model achieves better image reconstruction than earlier image-only stages while maintaining strong semantic understanding.

ATOKEN demonstrates significant advances in both scalability and performance. The model natively processes arbitrary resolutions and time durations, and accelerates inference through KV-caching mechanisms. To validate its effectiveness, we conduct comprehensive evaluations across three dimensions: reconstruction quality, semantic understanding, and downstream applications. These experiments confirm that ATOKEN achieves competitive or state-of-the-art performance across all modalities while maintaining computational efficiency.

## 2 BACKGROUND

Visual tokenization transforms raw visual data into compact representations for understanding and generation tasks. However, existing approaches remain fragmented across modalities and objectives, lacking the versatility of language models. To address space constraints, we provide a comprehensive overview of visual tokenization approaches organized along three critical dimensions in Table 6 (Appendix), while extensive related work discussion can be found in Section A (Appendix).

**Task Specialization.** Current visual tokenizers fall into two distinct categories: reconstruction methods (SD-VAE (Rombach et al., 2022), VQGAN (Esser et al., 2020), GigaTok (Xiong et al., 2025), Cosmos (Agarwal et al., 2025)) excel at compression for generation but cannot extract se-

mantic features; understanding encoders (CLIP (Radford et al., 2021), SigLIP2 (Tschannen et al., 2025), VideoPrism (Zhao et al., 2024)) produce rich semantics but cannot reconstruct content. Only VILA-U (Wu et al., 2024c) and UniTok (Ma et al., 2025) attempt both, limited to images. This divide prevents models that excel at both generation and understanding.

**Modality Fragmentation.** Beyond task specialization, visual tokenizers are limited to specific modalities. While most video tokenizers naturally handle images as single-frame videos (*e.g.*, TAE (Polyak et al., 2024), Hunyuan (Kong et al., 2024)), they cannot process 3D data. Conversely, 3D tokenizers like Trellis-SLAT (Xiang et al., 2024) are restricted to 3D-only data, unable to leverage the massive image and video data for pretraining. Understanding tasks face similar constraints: image encoders process videos frame-by-frame without temporal compression, while dedicated video encoders (Zhao et al., 2024; Wang et al., 2022b) lack image-specific optimizations.

**Architectural Trade-offs.** Key design trade-offs emerge across methods: *(1) Architecture:* Understanding encoders use transformers while reconstruction tokenizers favor convolutions (SD-VAE (Rombach et al., 2022)), with recent hybrid (GigaTok (Xiong et al., 2025)) and pure transformer (Vi-Tok (Hansen-Estruch et al., 2025)) approaches, the latter suffering from adversarial training instabilities. *(2) Token representation:* Methods choose discrete tokens for LLM compatibility (VQGAN (Esser et al., 2020)) or continuous tokens for reconstruction quality (TAE (Polyak et al., 2024)), with few supporting both. *(3) Resolution handling:* Convolutions naturally handle arbitrary resolutions, while only SigLIP2 (Tschannen et al., 2025) among transformers supports native resolution. *(4) Training objectives:* GAN-based training dominates reconstruction tokenizers despite instabilities.

## 3 MODEL

This section describes ATOKEN's architecture and training. We present our unified 4D representation for all modalities (Section 3.1), the transformer architecture processing these representations (Section 3.2), adversarial-free training objectives (Section 3.3), and a progressive curriculum for multimodal learning (Section 3.4). Detailed training recipes and implementation are in Section B.

### 3.1 UNIFIED LATENT REPRESENTATION

**Unified Modalities – Image, Video and 3D.** Our central insight is that all visual modalities can be represented within a shared 4D space. As illustrated in Figure 2, we process each modality through space-time patchification to produce sets of feature-coordinate pairs:

$$\boldsymbol{z} = \{(\boldsymbol{z}_i, \boldsymbol{p}_i)\}_{i=1}^{L}, \quad \boldsymbol{z}_i \in \mathbb{R}^C, \quad \boldsymbol{p}_i \in \{0, 1, \dots, N-1\}^4 \tag{1}$$

where $\boldsymbol{z}_i$ represents the latent feature at position $\boldsymbol{p}_i = [\mathrm{t}, \mathrm{x}, \mathrm{y}, \mathrm{z}]$ in 4D space (temporal and spatial coordinates), with $N$ defining the resolution along each axis and $L$ the number of active locations.

This sparse representation unifies all modalities by activating only their relevant dimensions: images occupy the $(\mathrm{x}, \mathrm{y})$ plane at $\mathrm{t} = \mathrm{z} = 0$, videos extend along the temporal axis with $\mathrm{z} = 0$, and 3D assets as surface voxels in $(\mathrm{x}, \mathrm{y}, \mathrm{z})$ space with $\mathrm{t} = 0$. For 3D assets, we adapt Trellis-SLAT (Xiang et al., 2024) by rendering multi-view images from spherically sampled cameras, applying our unified patchification, then aggregating features into voxel space (detailed in Section 3.2). This approach enables a single encoder $\mathcal{E}$ to process all modalities without architectural modifications.

**Unified Tasks – Reconstruction and Understanding.** From the unified structured latents $\boldsymbol{z} = \{(\boldsymbol{z}_i, \boldsymbol{p}_i)\}$, we extract representations for both reconstruction and understanding through complementary projections. For reconstruction, we project each latent to a lower-dimensional space $\boldsymbol{z}^r = \boldsymbol{W}_r(\boldsymbol{z})$ with KL regularization (Rombach et al., 2022), optionally applying FSQ (Mentzer et al., 2023) for discrete codes $\tilde{\boldsymbol{z}}^r = \mathrm{FSQ}(\boldsymbol{z}^r)$. The decoder $\mathcal{D}_\theta$ then reconstructs the input from these latents. For understanding, we aggregate latents via attention pooling (Radford et al., 2021; Tschannen et al., 2025) into a global representation $\bar{\boldsymbol{z}}$, which is projected to $\boldsymbol{z}^s = \boldsymbol{W}_s(\bar{\boldsymbol{z}})$ for alignment with text embeddings. This dual projection design allows joint optimization without architectural duplication – the same encoded features $\boldsymbol{z}$ support both pixel-level reconstruction through individual latents and semantic understanding through their aggregation.

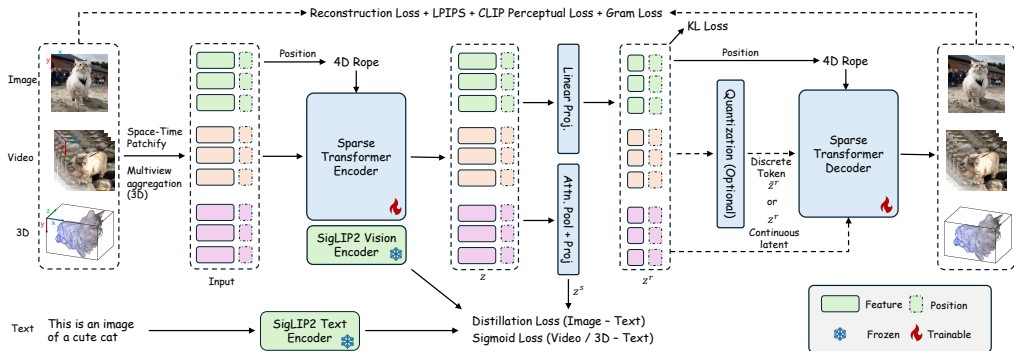

Figure 2: **Overview of our method.** All modalities undergo unified space-time patchification and encoding into sparse 4D latents, which support both reconstruction through modality-specific decoders and understanding through attention pooling and text alignment. The architecture jointly optimizes reconstruction and understanding losses, maintaining sparse structured representations throughout for efficient multimodal processing.

## 3.2 TRANSFORMER BASED ARCHITECTURE

**Unified Space-Time Patch Embedding.** We employ a unified patchification scheme that enables all modalities to share the same encoder. Given an input $x \in \mathbb{R}^{T \times H \times W \times 3}$, we partition it into non-overlapping space-time patches of size $t \times p \times p$. For images ($T = 1$), we apply temporal zero-padding to create $t$-frame patches, ensuring consistent dimensions across modalities. Videos are directly partitioned along both spatial and temporal dimensions.

For 3D assets, we adapt Trellis-SLAT (Xiang et al., 2024) to our unified pipeline. As shown in Figure 6 in the Appendix, we render multi-view images from spherically sampled cameras and apply our standard space-time patchification. Each voxel in a $64^3$ grid is back-projected to gather and average patch features from relevant views. Unlike Xiang et al. (2024), which uses DINOv2 features, we achieve comparable quality using our unified patch representation.

**Sparse Transformer Encoder and Decoder.** We employ a unified transformer architecture for both encoder and decoder, as illustrated in Figure 2. Both components process sparse structured representations – sets of feature-position pairs rather than dense grids – enabling efficient handling of all modalities with native support for arbitrary resolutions and temporal lengths.

Our encoder $\mathcal{E}$ extends the pretrained SigLIP2 vision tower (Tschannen et al., 2025) from 2D images to 4D representations through two modifications. First, we generalize patch embedding to space-time blocks of size $t \times p \times p$, with zero-initialized temporal weights preserving the original image features. Second, we augment SigLIP2's learnable 2D position embeddings with 4D RoPE (Lu et al., 2024a) applied in every attention layer, providing relative position awareness across $(\mathrm{t}, \mathrm{x}, \mathrm{y}, \mathrm{z})$ dimensions. This design maintains SigLIP2's semantic priors and resolution flexibility while enabling unified processing across modalities.

The decoder $\mathcal{D}$ shares the encoder's transformer architecture but is trained from scratch for reconstruction. It maps structured latents back to visual outputs through task-specific heads. For images and videos, we decode directly to pixel space:

$$\mathcal{D}_{\mathrm{P}} : \{(z_i, p_i)\}_{i=1}^{L} \rightarrow x \in \mathbb{R}^{T \times H \times W \times 3} \qquad (2)$$

treating images as single-frame videos ($T = 1$) and discarding temporal padding following Polyak et al. (2024). For 3D assets, we first decode to pixel-space features, then apply an additional layer to generate Gaussian splatting parameters for efficient rendering:

$$\mathcal{D}_{\mathrm{GS}} : \{(z_i, p_i)\}_{i=1}^{L} \rightarrow \{\{(o_i^k, c_i^k, s_i^k, \alpha_i^k, r_i^k)\}_{k=1}^{K}\}_{i=1}^{L} \qquad (3)$$

where each location generates $K$ Gaussians with parameters: position offset $o$, color $c$, scale $s$, opacity $\alpha$, and rotation $r$. Following Xiang et al. (2024), we constrain Gaussian positions to remain near their source voxels using $x_i^k = p_i + \texttt{tanh}(o_i^k)$, ensuring local feature coherence.

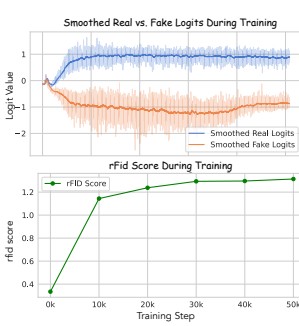 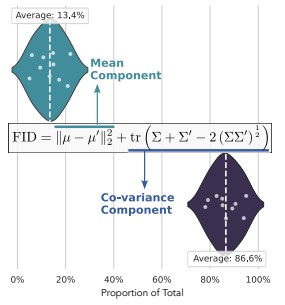 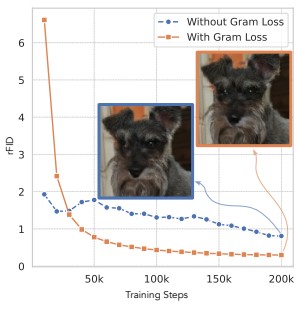

(a) GAN training instability     (b) Decomposition of rFID.     (c) Gram loss efficiency

Figure 3: **Adversarial-free training with Gram loss achieves stable, high-fidelity reconstruction.** (a) GAN training fails as the discriminator overpowers the generator, degrading rFID. (b) rFID decomposition shows $\approx 86.6\%$ of error stems from covariance (texture/style) vs. $\approx 13.4\%$ from mean. (c) Gram loss directly optimizes second-order statistics without adversarial training, achieving superior and stable rFID.

## 3.3 TRAINING OBJECTIVES

We jointly optimize for reconstruction and understanding through an adversarial-free training loss:

$$\mathcal{L} = \lambda_{\text{rec}}\mathcal{L}_{\text{rec}} + \lambda_{\text{sem}}\mathcal{L}_{\text{sem}} + \lambda_{\text{KL}}\mathcal{L}_{\text{KL}}, \tag{4}$$

where $\mathcal{L}_{\text{KL}}$ is the KL regularization on projected reconstruction latents $\boldsymbol{z}^r$, with weights $\lambda_{\text{rec}}$, $\lambda_{\text{sem}}$, $\lambda_{\text{KL}}$. We achieve state-of-the-art reconstruction without adversarial training, which is unstable at scale (Wu et al., 2025a) and incompatible with sparse 3D representations.

**Reconstruction Loss.** While GANs (Goodfellow et al., 2014) are standard for visual tokenizers, we found them unsuitable for our transformer architecture. Figure 3(a) shows the discriminator rapidly dominates the generator, causing mode collapse and degraded reconstruction quality. To develop an alternative, we analyzed the reconstruction error by decomposing rFID into mean and covariance components (Figure 3(b)). The covariance component – capturing second-order statistics like texture and style – dominates at $\approx 86.6\%$, while the mean contributes only 13.4%. This motivated adopting Gram matrix loss (Gatys et al., 2016), which directly optimizes feature covariance without adversarial training by computing the Gram matrix $G(F) = FF^\top$ for feature maps from different layers. As shown in Figure 3(c), this achieves superior performance throughout training.

For images, we combine four complementary loss components:

$$\mathcal{L}_{\text{rec}}^{\text{I}} = \lambda_1\mathcal{L}_1 + \lambda_{\text{LPIPS}}\mathcal{L}_{\text{LPIPS}} + \lambda_{\text{GRAM}}\mathcal{L}_{\text{GRAM}} + \lambda_{\text{CLIP}}\mathcal{L}_{\text{CLIP}}, \tag{5}$$

where $\mathcal{L}_1 = \|\boldsymbol{x} - \hat{\boldsymbol{x}}\|_1$ provides pixel supervision, $\mathcal{L}_{\text{LPIPS}}$ (Zhang et al., 2018) measures perceptual similarity, $\mathcal{L}_{\text{GRAM}}$ captures texture, and $\mathcal{L}_{\text{CLIP}}$ enforces semantic consistency. For video and 3D assets, we use $\mathcal{L}_{\text{rec}}^{\text{V/3D}} = \mathcal{L}_1$ for efficiency, relying on cross-modal transfer from images for details:

$$\mathcal{L} = \lambda_{\text{rec}}\mathcal{L}_{\text{rec}} + \lambda_{\text{sem}}\mathcal{L}_{\text{sem}} + \lambda_{\text{KL}}\mathcal{L}_{\text{KL}}, \tag{6}$$

where $\mathcal{L}_{\text{KL}}$ is the KL regularization term applied to the projected reconstruction latents $\boldsymbol{z}^r$, with $\lambda_{\text{rec}}$, $\lambda_{\text{sem}}$ and $\lambda_{\text{KL}}$ balancing components. Notably, we achieve state-of-the-art reconstruction quality without adversarial training, which has been observed to be unstable when scaling (Wu et al., 2025a) and incompatible with our sparse 3D representations.

**Semantic Loss.** We align visual representations $\boldsymbol{z}^s$ with text embeddings through modality-specific objectives. For images, we distill knowledge from the frozen SigLIP2 vision encoder by minimizing the KL divergence between temperature-scaled vision-text similarity distributions:

$$\mathcal{L}_{\text{sem}}^{\text{I}} = \text{KL}\left(\text{softmax}(\tau^{-1}\boldsymbol{s}^{\text{teacher}}) \,\|\, \text{softmax}(\tau^{-1}\boldsymbol{s}^{\text{student}})\right), \tag{7}$$

where $\boldsymbol{s}^{\text{teacher}}$ and $\boldsymbol{s}^{\text{student}}$ are vision-text similarity scores from frozen SigLIP2 and our model respectively, both paired with the same frozen text encoder, and $\tau$ is the temperature parameter. For videos and 3D, we directly optimize alignment using the sigmoid loss from SigLIP (Zhai et al., 2023), which proves more stable for the smaller batch sizes typical in these domains. This dual strategy preserves pretrained image semantics while enabling efficient learning for new modalities.

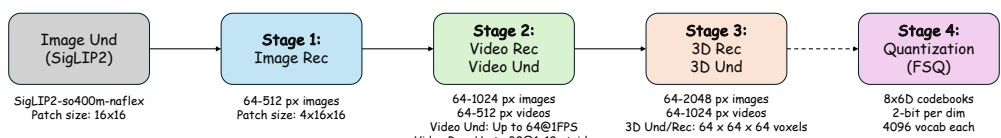

Figure 4: **Progressive training curriculum of AToken.** Our model starts from SigLIP2 image understanding and progressively adds: (1) image reconstruction, (2) video capabilities with temporal modeling, (3) 3D understanding with expanded resolutions, and optionally (4) discrete tokenization via FSQ. Each box shows the new capabilities introduced at that stage, along with supported resolutions, patch sizes, and sampling strategies.

## 3.4 TRAINING STRATEGY

Our training employs a four-stage curriculum (Figure 4) that builds from image foundations to video dynamics to 3D geometry, with optional discrete quantization. Starting from the pretrained SigLIP2 vision encoder, we gradually introduce more complex objectives and modalities while using gradient accumulation to balance image-text distillation with reconstruction, video-text alignment, and 3D-text alignment across all stages. This ensures semantic alignment is preserved as reconstruction capabilities expand through round-robin sampling.

**Stage 1: Image Foundation.** Starting from pretrained SigLIP2, we establish core visual representations by adding image reconstruction capabilities with 32 latent dimensions (Yao & Wang, 2025). Training uses variable resolution sampling from 64 to 512 pixels.

**Stage 2: Video Dynamics.** We extend to temporal sequences, expanding latent dimensions to 48 for motion complexity (Seawead et al., 2025). Resolution increases to 1024 for images and 512 for videos. We employ temporal tiling with adaptive sampling and KV-caching (Figure 7 in Appendix) to eliminate redundant computation.

**Stage 3: 3D Geometry.** We incorporate 3D assets as $64^3$ voxel grids, using Gaussian splatting for reconstruction and attention pooling for understanding. Resolution further increases to 2048 for images and 1024 for videos. Joint optimization across modalities prevents catastrophic forgetting while leveraging cross-modal learning.

**Stage 4: Discrete Tokenization.** Optionally, we add FSQ quantization (Mentzer et al., 2023), partitioning 48-dimensional latents into 8 discrete tokens from 4096-entry codebooks, enabling compatibility with discrete generative models across all visual domains.

See Section B.3 for complete training configurations and Section B.4 for implementation details.

## 4 RESULTS

We evaluate ATOKEN as the first visual tokenizer to achieve reconstruction and understanding across images, videos, and 3D assets. Our evaluation demonstrates that unified tokenization achieves competitive performance across all modalities (Section 4.1), seamlessly integrates into existing understanding pipelines (Section 4.3), enables high-quality generation without architectural changes (Section 4.4), and reveals critical insights about model scaling and cross-modal benefits (Section 4.2). In the Appendix, Section C reveals progressive improvements with detailed per-modality evaluations, and Section D validates versatility in video generation and 3D synthesis applications.

## 4.1 UNIFIED TOKENIZER EVALUATION

Table 1 compares visual tokenizers across modalities using ImageNet (Deng et al., 2009) (reconstruction: rFID; understanding: zero-shot accuracy), TokenBench (Agarwal et al., 2025), MSR-VTT (Xu et al., 2016) for video, and Toys4k (Stojanov et al., 2021a) for 3D. Existing approaches fall into three limited categories: reconstruction-only tokenizers (SD-VAE (Rombach et al., 2022), Hunyuan (Kong et al., 2024), Trellis-SLAT (Xiang et al., 2024)) excel at generation but lack semantics; understanding-only encoders (SigLIP2 (Tschannen et al., 2025), VideoPrism (Zhao et al., 2024)) provide semantics but cannot reconstruct; recent unified attempts (SeTok (Wu et al., 2024b), UniTok (Ma et al., 2025)) combine both but remain image-only.

Table 1: **Performance comparison of visual tokenizers across modalities.** We evaluate on ImageNet for image reconstruction and zero-shot classification, TokenBench for video reconstruction with MSR-VTT, and Toys4k for 3D reconstruction and classification. Discrete tokenizers are indicated with gray shading.

| Method | Comp. Ratio | Latent Channels | Token Type | Image | | | Video | | | 3D | | |
|---|---|---|---|---|---|---|---|---|---|---|---|---|
| | | | | PSNR↑ | rFID↓ | Acc.↑ | PSNR↑ | rFVD↓ | R@1↑ | PSNR↑ | LPIPS↓ | Acc.↑ |
| *Reconstruction Only* | | | | | | | | | | | | |
| SD-VAE | (1, 8, 8) | 4 | VAE | 26.26 | 0.61 | - | - | - | - | - | - | - |
| FLUX.1 [dev] | (1, 8, 8) | 16 | VAE | 32.86 | **0.18** | - | - | - | - | - | - | - |
| VA-VAE | (1, 16, 16) | 32 | VAE | 27.70 | 0.28 | - | - | - | - | - | - | - |
| GigaTok-XL-XXL | (1, 16, 16) | 8 | VQ | 22.42 | 0.80 | - | - | - | - | - | - | - |
| Cosmos-0.1-CV8×8 | (4, 8, 8) | 16 | AE | 30.11 | 7.55 | - | 34.33 | 8.34 | - | - | - | - |
| OmniTokenizer† | (4, 8, 8) | 8 | VAE | 26.74 | 1.02 | - | 19.39 | 173.48 | - | - | - | - |
| Hunyuan | (4, 8, 8) | 16 | VAE | **33.32** | 0.67 | - | 36.37 | 3.78 | - | - | - | - |
| Wan2.2 | (4, 16, 16) | 48 | VAE | 31.25 | 0.75 | - | **36.39** | 3.19 | - | - | - | - |
| OmniTokenizer† | (4, 8, 8) | 8 | VQ | 24.69 | 1.41 | - | 19.89 | 202.46 | - | - | - | - |
| Cosmos-0.1-DV8×8 | (4, 8, 8) | 6 | FSQ | 26.34 | 7.86 | - | 31.42 | 25.94 | - | - | - | - |
| Trellis-SLAT | - | 8 | VAE | - | - | - | - | - | - | 26.97 | **0.054** | - |
| *Understanding Only* | | | | | | | | | | | | |
| SigLIP2-So/16 | (1, 16, 16) | - | - | - | - | 83.4 | - | - | 41.9 | - | - | - |
| PE$_{core}$L | (1, 14, 14) | - | - | - | - | **83.5** | - | - | **50.3** | - | - | - |
| *Reconstruction & Understanding* | | | | | | | | | | | | |
| VILA-U | (1, 16, 16) | 16 | RQ | 22.24 | 4.23 | 78.0 | - | - | - | - | - | - |
| UniTok | (1, 16, 16) | 64 | MCQ | 25.34 | **0.36** | 78.6 | - | - | - | - | - | - |
| ATOKEN-So/D | (4, 16, 16) | 48 | FSQ | **27.00** | 0.38 | 82.2 | **33.12** | **22.16** | 40.3 | 28.17 | 0.063 | **91.3** |
| ATOKEN-So/C | (4, 16, 16) | 48 | VAE | 29.72 | 0.21 | 82.2 | 36.07 | **3.01** | 40.2 | 28.28 | 0.062 | 90.9 |

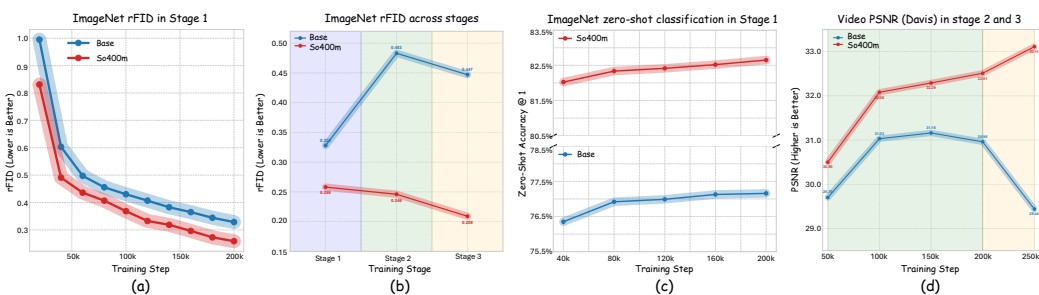

Figure 5: **Architectural scaling comparison: Base vs. So400m models.** (a) ImageNet rFID during Stage 1 training. (b) ImageNet rFID across training stages. (c) ImageNet zero-shot classification accuracy in Stage 1. (d) Video PSNR on DAVIS in Stages 2 and 3. The So400m model maintains or improves performance across all stages, while the Base model shows significant degradation when extending beyond single-modality training, indicating that sufficient model capacity is critical for successful multimodal visual tokenization.

ATOKEN-So/C breaks these boundaries as the first unified tokenizer across all modalities, achieving 0.21 rFID with 82.2% ImageNet accuracy (vs. UniTok's 0.36 rFID and 78.6%), while extending to video (3.01 rFVD, 40.2% R@1) and 3D (28.28 PSNR, 90.9% accuracy), matching specialized methods like Wan2.2 (Wan et al., 2025) and Trellis-SLAT. Our discrete variant (ATOKEN-So/D) maintains competitive performance, pioneering discrete tokenization across all modalities. Detailed evaluations with comprehensive baselines are in Appendix C: image reconstruction and zero-shot benchmarks (Section C.1), video reconstruction and retrieval (Section C.2), 3D reconstruction and classification (Section C.3), with qualitative visualizations (Figures 9 to 11).

## 4.2 SCALING AND CROSS-MODAL BENEFITS

To investigate the scaling property of the visual tokenizer, we compare our So400m model with a smaller Base variant following identical training procedures. The Base model initializes from SigLIP-Base-patch16-naflex (Tschannen et al., 2025), comprising 12 transformer blocks with hidden dimension $d = 768$ and 12 attention heads for both encoder and decoder, yielding approximately 192M parameters compared to So400m's 800M.

Table 2: **Image understanding comparison across multimodal LLMs.** Evaluation of SlowFast-LLaVA-1.5 with frozen ATOKEN-So/C vision encoder versus Oryx-ViT and other state-of-the-art MLLMs.

| Multimodal LLM | Vision Encoder | # Input Pixels | General & Knowledge | | | | | TextRich | |
|---|---|---|---|---|---|---|---|---|---|
| | | | RW-QA (test) | AI2D (test) | SQA (test) | MMMU (val) | MathV (testmini) | OCRBench (test) | TextVQA (val) |
| *1B Model Comparison* | | | | | | | | | |
| MolmoE-1B | MetaCLIP | 4.10M | 60.4 | 86.4 | - | 34.9 | 34.0 | - | 78.8 |
| SlowFast-LLaVA-1.5-1B | Oryx-ViT | 2.36M | 59.2 | 72.8 | 87.7 | 40.5 | 51.0 | 70.0 | 71.3 |
| SlowFast-LLaVA-1.5-1B | ATOKEN-So/C | 2.36M | 60.1 | 74.2 | 88.7 | 40.6 | 52.5 | 67.6 | 72.5 |
| *3B Model Comparison* | | | | | | | | | |
| MM1.5-3B | CLIP | 4.52M | 56.9 | 65.7 | 85.8 | 37.1 | 44.4 | 65.7 | 76.5 |
| SlowFast-LLaVA-1.5-3B | Oryx-ViT | 2.36M | 63.4 | 77.0 | 90.3 | 44.7 | 58.6 | 73.4 | 73.0 |
| SlowFast-LLaVA-1.5-3B | ATOKEN-So/C | 2.36M | 64.3 | 79.1 | 89.7 | 45.7 | 58.4 | 73.3 | 72.8 |
| *7B Model Comparison* | | | | | | | | | |
| Oryx1.5-7B | Oryx-ViT | 2.36M | - | 79.7 | - | 47.1 | - | 71.3 | 75.7 |
| InternVL2.5-8B | InternViT | 9.63M | 70.1 | 84.5 | - | 56.0 | 64.4 | - | 79.1 |
| Qwen2-VL-7B | DFN | - | 70.1 | 83.0 | - | 54.1 | 58.2 | - | 84.3 |
| SlowFast-LLaVA-1.5-7B | Oryx-ViT | 2.36M | 67.5 | 80.4 | 91.1 | 49.0 | 62.5 | 76.4 | 76.4 |
| SlowFast-LLaVA-1.5-7B | ATOKEN-So/C | 2.36M | 68.8 | 81.2 | 92.1 | 48.7 | 61.2 | 74.5 | 77.7 |

Table 3: **Video understanding performance on multimodal LLMs.** Evaluation of SlowFast-LLaVA-1.5 with frozen ATOKEN-So/C vision encoder versus Oryx-ViT and other video MLLMs.

| Multimodal LLM | Vision Encoder | # Input Tokens | General VideoQA | | | Long-Form Video Understanding | | |
|---|---|---|---|---|---|---|---|---|
| | | | VideoMME (w/o sub) | PercepTest (val) | NExT-QA (test) | LongVideoBench (val) | MLVU (m-avg) | LVBench (avg) |
| *1B Model Comparison* | | | | | | | | |
| Qwen2-VL-2B | DFN | 16K | 55.6 | 53.9 | 77.2 | 48.7 | 62.7 | 39.4 |
| SlowFast-LLaVA-1.5-1B | Oryx-ViT | 9K | 56.6 | 61.9 | 76.7 | 54.3 | 64.3 | 39.7 |
| SlowFast-LLaVA-1.5-1B | ATOKEN-So/C | 9K | 56.7 | 63.9 | 74.8 | 55.1 | 64.7 | 41.1 |
| *3B Model Comparison* | | | | | | | | |
| Apollo-3B | SigLIP | 3K | 58.4 | 65.0 | - | 55.1 | 68.7 | - |
| SF-LLaVA-1.5-3B | Oryx-ViT | 9K | 60.8 | 65.8 | 80.8 | 57.2 | 68.8 | 43.3 |
| SF-LLaVA-1.5-3B | ATOKEN-So/C | 9K | 60.4 | 66.0 | 80.8 | 57.2 | 66.7 | 41.3 |
| *7B Model Comparison* | | | | | | | | |
| InternVL2.5-8B | InternViT | 16K | 64.2 | - | 85.0 | 60.0 | 69.0 | 43.2 |
| Qwen2-VL-7B | DFN | 16K | 63.3 | 62.3 | 81.2 | 55.6 | 69.8 | 44.7 |
| SlowFast-LLaVA-1.5-7B | Oryx-ViT | 9K | 63.9 | 69.6 | 83.3 | 62.5 | 71.5 | 45.3 |
| SlowFast-LLaVA-1.5-7B | ATOKEN-So/C | 9K | 64.5 | 70.3 | 83.7 | 60.6 | 69.8 | 44.8 |

As shown in Figure 5, both models achieve reasonable single-modal performance in Stage 1, with So400m outperforming Base (0.258 vs 0.323 rFID, 82.7% vs 77.2% accuracy). However, the Base model suffers severe degradation when expanding to videos, with ImageNet rFID degrading 49% (0.323→0.483) and video PSNR declining across stages. In contrast, So400m improves continuously – ImageNet rFID enhances 19% (0.258→0.209) while video PSNR rises from 32.51 to 33.11. This scaling analysis reveals that multimodal tokenization has a capacity requirement: small models suffer from interference while large models benefit from cross-modal learning. Additional ablations and extensive reconstruction visualizations across all modalities are provided in Section C.4.

## 4.3 MULTIMODAL LLMS

To validate ATOKEN's effectiveness for vision-language understanding, we integrate it into SlowFast-LLaVA-1.5 (Xu et al., 2025), replacing the Oryx-ViT (Liu et al., 2024b) vision encoder with ATOKEN-So/C while freezing ATOKEN parameters during training.

**Image Understanding.** Table 2 shows results on 7 standard benchmarks including RW-QA, AI2D (Kembhavi et al., 2016), SQA (Lu et al., 2022b), MMMU (Yue et al., 2024), MathVISTA (Lu et al., 2024b), OCRBench (Liu et al., 2024a), and TextVQA (Singh et al., 2019). SlowFast-LLaVA-1.5 with ATOKEN outperforms Oryx-ViT across model scales, with the 7B model achieving gains of 1.3% on RW-QA, 1.0% on SQA, and 1.3% on TextVQA. The 3B model achieves superior results on almost all benchmarks, demonstrating strong generalization ability.

**Video Understanding.** Table 3 covers video tasks including Video-MME (Fu et al., 2024), PercepTest (Pătrăucean et al., 2023), NExT-QA (Xiao et al., 2021), and long-video benchmarks LongVideoBench (Wu et al., 2025b), MLVU (Zhou et al., 2024b), and LVBench (Wang et al.,

Table 4: **Continuous tokenizers** on ImageNet.

| Tokenizer | CFG | gFID↓ | IS↑ | Pre.↑ | Rec.↑ |
|---|---|---|---|---|---|
| DiT | 1.5 | 2.27 | 278.2 | **0.83** | 0.57 |
| REPA | 1.35 | 1.42 | **305.7** | 0.80 | **0.65** |
| VAVAE | 6.7† | **1.35** | 295.3 | 0.79 | **0.65** |
| ATOKEN-So/C | | | | | |
| Stage 1 | 1.5 | 1.62 | 253.3 | 0.78 | 0.63 |
| Stage 2 | 1.65 | 1.88 | 231.1 | 0.80 | 0.60 |
| Stage 3 | 1.65 | 1.56 | 260.0 | 0.79 | 0.63 |

Table 5: **Discrete tokenizers** on ImageNet.

| Tokenizer | CFG | gFID↓ | IS↑ | Pre.↑ |
|---|---|---|---|---|
| LFQ | - | 1.91 | **324.3** | - |
| TikTok-L | - | 6.18 | 182.1 | 0.80 |
| VQGAN | 1.75 | 2.34 | 253.9 | 0.81 |
| UniTok | 1 | 2.51 | 216.7 | **0.82** |
| TokenBridge | 3.1 | **1.76** | 294.8 | 0.80 |
| ATOKEN-So/D | 3.1 | 2.23 | 274.5 | 0.79 |

2024c). ATOKEN excels at smaller scales, with the 1.5B model achieving state-of-the-art performance on most benchmarks. It provides strong gains on general video QA, achieving 64.5% on VideoMME and 70.3% on PercepTest with 7B models. While Oryx-ViT shows advantages on long-form understanding (particularly MLVU), likely due to its video-specific design, ATOKEN demonstrates competitive unified performance across modalities.

## 4.4 IMAGE GENERATION WITH CONTINUOUS & DISCRETE TOKENS

**Continuous Tokens.** We evaluate continuous token generation using Lightning-DiT (Yao & Wang, 2025), comparing against diffusion methods (DiT (Peebles & Xie, 2022), SiT (Ma et al., 2024a)) and reconstruction-specialized approaches (REPA (Yu et al., 2024b), VAVAE (Yao & Wang, 2025)). For fair comparison with VAVAE, we use identical training code, adapting only for ATOKEN's 48-dimensional latents (vs. 32). Following Lightning-DiT protocols, we apply CFG scale 1.65 across all channels. As shown in Table 4, ATOKEN-So/C achieves 1.56 gFID, competitive with VAVAE (1.35) and REPA (1.42) despite optimizing for multiple modalities. The So model improves from Stage 2 to Stage 3 (1.88→1.56 gFID), suggesting multimodal training can enhance generation quality when given sufficient capacity.

**Discrete Tokens.** We integrate ATOKEN-So/D into the TokenBridge (Wang et al., 2025) autoregressive framework, replacing only the tokenizer. Unlike TokenBridge's 16 dimensions with 8-level vocabularies, ATOKEN-So/D uses 8 dimensions with 4096-level vocabularies – a more challenging configuration requiring modeling of larger discrete spaces. As shown in Table 5, ATOKEN-So/D achieves 2.23 gFID, outperforming UniTok (2.51 gFID), the only other unified visual tokenizer. While TokenBridge achieves a lower gFID (1.76), this gap is expected given our larger vocabulary size (4096 vs. 8), demonstrating that multimodal capabilities need not compromise generation quality.

**Extended Generative Applications.** Our unified tokens enable diverse downstream tasks beyond images. For text-to-video (Section D.1), ATOKEN-So/C achieves 78.46% VBench score, matching specialized tokenizers (Wan2.1: 78.60%, Hunyuan 78.02%). For image-to-3D synthesis (Section D.2), we successfully generate 3D assets from single images, though our 48-dimensional latents require further optimization versus task-specific 8-channel approaches. These results validate unified tokenization as a foundation for multimodal generation (samples: Figure 12–Figure 14).

## 5 DISCUSSION AND CONCLUSION

The effectiveness of ATOKEN across diverse modalities and tasks suggests new opportunities: visual tokenization can achieve the same unification that transformed language modeling. Our single framework achieves both high-fidelity reconstruction and semantic understanding across images, videos, and 3D assets. This integration became possible through the combination of our sparse 4D representation, transformer-based architecture, adversarial-free training strategy, and progressive multimodal curriculum. Due to limited computational resources, we could only test ATOKENon separate downstream tasks. Building the comprehensive omnimodel that would demonstrate ATOKEN's full potential remains as future work. Looking forward, ATOKEN opens paths for visual foundation models to follow language modeling's trajectory toward true generalization. We hope this work sheds light on the next-generation multimodal AI systems built upon unified visual tokenization.

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
