# OpenReview forum: "AToken: A Unified Tokenizer for Vision"
_ICLR.cc/2026/Conference — ICLR 2026 Conference Withdrawn Submission_

### Official Review · Reviewer_Kup7 · 2025-10-29

**Soundness:** 4
**Presentation:** 3
**Contribution:** 3
**Rating:** 8
**Confidence:** 5

**Summary:**

This paper introduces AToken, a unified visual tokenizer arcross images, videos and 3D asset. By incorporating 4D positional embedding, space-time patch embedding, and a sparse transformer encoder, it enables different types of visual inputs to share a unified representation space. From a technical perspective, the paper introduces a Gram loss to replace the traditional GAN loss for more stable training, along with a well-designed training recipe that successfully scales to various input types. Extensive experiments demonstrate that AToken achieves excellent performance in both reconstruction and understanding tasks.

**Strengths:**

- This paper attempts to address a fundamental and important issue — how to unify different types of visual signals — and proposes a relatively simple yet effective solution (4D positional embedding, space-time patch embedding, and an image-pretrained sparse transformer encoder).
- This paper introduces a Gram loss that directly optimizes based on second-order statistics to replace the GAN loss, which is both convincing and technically solid.
- The paper conducts extensive experiments and ablation studies, clearly demonstrating the impact of different training methods and stages on performance, and performs evaluations across various downstream tasks.

**Weaknesses:**

- Some implementation details are missing or difficult to locate.

**Questions:**

- Why is it necessary to use an additional distillation loss for images instead of using only the sigmoid loss, as done for videos and 3D assets? After all, distillation loss requires an extra frozen image encoder. How large is the performance gap between the two methods?
- The implementation details of the CLIP perceptual loss are unclear.
- The arrangement of the formulas is somewhat confusing. For example, Equation (4) and Equation (6) are identical.

---

### Official Review · Reviewer_ZxHw · 2025-10-30

**Soundness:** 3
**Presentation:** 2
**Contribution:** 2
**Rating:** 4
**Confidence:** 4

**Summary:**

This paper presents ATOKEN, a unified visual tokenizer that achieves both reconstruction and semantic understanding across images, videos, and 3D assets. ATOKEN employs a progressive training curriculum, gradually expanding its capability from single images to videos and then to 3D data. Experimental results demonstrate that ATOKEN achieves promising performance across these modalities.

**Strengths:**

- The paper is well written.

 - ATOKEN is a unified tokenizer capable of encoding images, videos, and 3D assets, demonstrating great potential in bridging multimodal representations.

**Weaknesses:**

- The generation results of the tokenizer are an important evaluation of its effectiveness. However, it seems that ATOKEN does not achieve state-of-the-art (SOTA) performance in generation. Moreover, ATOKEN contains 192M parameters, which is much larger than typical tokenizers. I believe this would make the training of generative models based on it more difficult, and I therefore have concerns about the tokenizer’s generative capability.

 - The experimental results in Table 1 show that the model achieves SOTA performance in understanding tasks, but its reconstruction ability still lags behind. I hope the authors can clarify this discrepancy.

 - The authors should also explain the changes in gFID after Stage 2 and Stage 3, as shown in the Table 4.

**Questions:**

NA

---

### Official Review · Reviewer_why4 · 2025-11-01

**Soundness:** 3
**Presentation:** 3
**Contribution:** 4
**Rating:** 8
**Confidence:** 5

**Summary:**

* This paper propose ATOKEN, a unified visual tokenizer, enables high-fidelity reconstruction and semantic understanding across images, videos, and 3D via a 4D latent space, transformer architecture, adversarial-free training, and progressive curriculum, achieving strong performance on benchmarks

* It addresses three long-standing limitations in existing visual representation systems: the fragmentation between reconstruction-focused and understanding-focused models, the restriction to single modalities, and the instability of transformer-based tokenizers during training.

* ATOKEN adopts several key technical designs: Unified 4D Latent Space,  A pure transformer with space-time patch embedding and 4D Rotary Position Embeddings (4D RoPE) ,Adversarial-Free Training Objective.

**Strengths:**

After reading this paper, I quite like it. I believe this paper makes considerable technical contributions and has certain innovations. Specifically:

1. It is the first visual tokenizer that unifies 2D & 3D reconstruction & generation, and achieves excellent performance on both understanding and generation-related benchmarks, with substantial workload and abundant experiments .

2. The 4D space design is quite reasonable, and a corresponding 4D RoPE is also designed. Using a transformer to achieve state-of-the-art (SOTA) performance—from what I know, the current reconstruction and generation effects of transformer-based tokenizers are inferior to those of CNN-based tokenizers .

3. It directly optimizes the texture/style errors in reconstruction through Gram matrix loss, replacing traditional GAN training. While ensuring reconstruction quality, it solves the training instability problem of large-scale visual transformers, and I think this optimization is very promising .

4. Many engineering designs are worthy of praise, such as supporting native resolution and conducting experimental verification for both discrete and continuous tokens.

**Weaknesses:**

Of course, I also think this paper has several shortcomings:

1. The reason why the transformer achieves state-of-the-art (SOTA) reconstruction performance for the image & video tokenizer is not clearly explained, nor is there a detailed ablation study to clarify the causes of this performance .

2. Since ATOKEN uses FSQ with a 48-dimensional latent space—where the 48-dimensional latents are partitioned into 8 groups of 6 dimensions, each quantized to 4 levels—and adopts the TokenBridge scheme, does this mean an additional autoregressive (AR) head is required? Additionally, will the FSQ design make it more difficult for the visual tokenizer to align with the language codebook, thereby hindering the formation of a unified codebook?

**Questions:**

The description of the Gram matrix loss is not clear enough. It is hoped that the authors will elaborate on its implementation and provide a detailed ablation study

---

### Official Review · Reviewer_hgrt · 2025-11-01

**Soundness:** 2
**Presentation:** 3
**Contribution:** 3
**Rating:** 6
**Confidence:** 3

**Summary:**

This paper proposes AToken, a unified tokenizer that bridges visual generation and visual understanding. Experiments demonstrate that AToken achieves superior performance on ImageNet image reconstruction, ImageNet zero-shot image classification, multimodal understanding and generation tasks.

**Strengths:**

1. AToken is the first work to explore a unified tokenizer that simultaneously supports single images, videos, and 3D data, representing a major step toward universal visual tokenization. This generality significantly broadens the applicability of discrete tokenizers beyond 2D imagery and establishes a strong foundation for multimodal generative models.
2. The model achieves competitive or superior reconstruction performance without relying on GAN-based objectives, instead combining perceptual and Gram-matrix losses to preserve fine-grained texture and global consistency. This makes AToken more stable and easier to train while maintaining visual fidelity comparable to adversarially trained tokenizers.
3. The paper presents comprehensive evaluations across multiple modalities and datasets, including image/video/3D reconstruction, multimodal understanding and generation.

**Weaknesses:**

1. Several important results are missing: 1) text-to-image generation results 2) multimodal LLM benchmarks using AToken-so/D
2. It would be better to show the ablation results using or not Gram loss for visual reconstruction
3. It would be better to do ablation studies on the depth of Sparse Encoder/Decoder, progressive training

**Questions:**

Please refer to the weakness section.

---

### Note · Authors · 2025-11-12

I have read and agree with the venue's withdrawal policy on behalf of myself and my co-authors.